# Increases of Phosphorylated Tau (Ser202/Thr205) in the Olfactory Regions Are Associated with Impaired EEG and Olfactory Behavior in Traumatic Brain Injury Mice

**DOI:** 10.3390/biomedicines10040865

**Published:** 2022-04-07

**Authors:** Younghyun Yoon, SuHyun Kim, YunHee Seol, Hyoenjoo Im, Uiyeol Park, Hio-Been Han, Jee Hyun Choi, Hoon Ryu

**Affiliations:** 1Department of Electrical Engineering and Computer Science, Vanderbilt University, Nashville, TN 37235, USA; yhyoon06@gmail.com; 2Center for Neuroscience, Brain Science Institute, Korea Institute of Science and Technology (KIST), Seoul 02792, Korea; shkimail@kist.re.kr (S.K.); yseol@kaist.ac.kr (Y.S.); imhj@standup.science (H.I.); uypark@kist.re.kr (U.P.); hiobeen.han@kaist.ac.kr (H.-B.H.); 3Program of Brain and Cognitive Engineering, Korea Advanced Institute of Science and Technology, Daejeon 34141, Korea; 4Neuroscience Program, Division of Bio-Medical Science & Technology, KIST School, University of Science and Technology, Seoul 02792, Korea; 5Boston University Alzheimer’s Disease Research Center, Department of Neurology, Boston University School of Medicine, Boston, MA 02130, USA

**Keywords:** traumatic brain injury (TBI), olfactory system, electroencephalogram (EEG), p-Tau, neuronal atrophy

## Abstract

Traumatic brain injury (TBI) leads to long-term cognitive impairments, with an increased risk for neurodegenerative and psychiatric disorders. Among these various impairments, olfactory dysfunction is one of the most common symptoms in TBI patients. However, there are very few studies that show the association between olfactory dysfunction and repetitive TBI. To investigate the effects of repetitive TBI on olfactory functioning and the related pathological neuronal injuries in mice, we applied a weight-drop model of TBI and performed neuropathological examinations and electroencephalography (EEG) in olfactory-bulb-associated areas. Through neuropathological examinations, we found significant increases of amyloid precursor protein (APP) and phosphorylated Tau (p-Tau) (S202/T205) in olfactory-bulb-associated areas. Neuronal atrophy in the lateral anterior olfactory nucleus (AOL), granule layer olfactory bulb (GrO), and dorsal tenia tecta (DTT) was also found to be correlated with p-Tau levels. However, there was no difference in the total Tau levels in the olfactory-bulb-associated areas of TBI mice. Electroencephalography (EEG) of repetitive TBI mouse models showed impaired spontaneous delta oscillation, as well as altered cross-frequency coupling between delta phase and amplitudes of the fast oscillations in the resting-state olfactory bulb. Furthermore, abnormal alterations in EEG band powers were observed during the olfactory oddball paradigm test. TBI also led to impairments of the olfactory-function-associated behaviors. This study provides evidence of behavioral, neuropathological, and physiological alterations in the mouse olfactory system caused by repetitive TBI. Together, p-Tau alterations and EEG impairments may serve as important biomarkers of olfactory-track-associated dysfunctions in repetitive TBI.

## 1. Introduction

Traumatic brain injury (TBI) accounts for approximately 1.7 million patients yearly in the United States and remains as a leading cause of morbidity worldwide [1,2,3]. Depending on the nature and severity of TBI, anatomical changes are highly variable and often result in long-term neurobehavioral and neuropathological sequelae [4,5,6]. Clinical manifestations may include neuropsychiatric disturbances, such as memory impairment, and difficulty sleeping, as well as mood or behavioral abnormalities, such as depression, aggression, and anxiety [7]. Out of the many pathological phenotypes, olfactory dysfunction is often an early signal for chronic neurodegenerative disorders, such as chronic traumatic encephalopathy (CTE), Alzheimer’s disease (AD), and Parkinson’s disease [5,6,8,9]. For instance, pathophysiological markers, such as α-synuclein and strongest p-Tau signal, have been linked to changes in the olfactory system and are well-known to be indicators of dementia risk. In the case of Parkinson’s disease, the olfactory bulb has been proposed as a possible induction site for Lewy pathology. Pathological changes were also observed in other areas of the olfactory system, such as the anterior olfactory nucleus (AON), cortical nucleus of the amygdala, piriform cortex, olfactory tubercle, entorhinal cortex, and orbitofrontal cortex [10].

While the associations between olfactory dysfunction and its pathological markers have been mainly observed in cases of moderate-to-severe TBI, in recent years, special interest has headed toward those of repetitive TBI [11,12,13,14,15,16]. However, a gap in TBI studies has been the use of a simple model system that accurately reads the physiological and behavioral changes of olfactory disruption after TBI events [6]. Recent studies show that advanced human imaging tools, such as MRI and CT scans, can detect pathology and prognostication in various brain regions of mild TBI patients [2,17]. The state-of-the art imaging methodology even can determine pathology and prognostication within 72 h after TBI, rather than 2 to 3 weeks. Furthermore, Callaham and Hinkebein found that 65% of TBI patients experienced anosmia, although most were unaware of their impaired olfaction [5]. Anosmia was also correlated with greater cognitive disability [5]. These subtle behavioral differences in the olfactory dysfunction show that a sensitive measuring tool is needed to detect TBI related injuries in human patients. Past TBI models have utilized invasive methods, such as controlled cortical injury (CCI), blast-induced TBI (bTBI), fluid percussion injury (FPI), and closed/head impact model of engineered rotational acceleration (CHIMERA). In this study, we produced TBI mice with repetitive impacts via weight drop on the head of mice to emulate human sub-TBI scenarios and examined the various neurological consequences of repetitive head injuries. Although the buried-food-seeking test is the most reliable olfactory assay in rodents so far [18,19], it is reported to be closely associated with other cognitive deficits, such as abnormalities in food motivation or elevated fear. For this reason, it is hard to detect and measure subtle olfactory impairments may remain undetected [20]. To overcome this weakness, we used electroencephalogram (EEG) to measure neuronal oscillations of mice during an olfactory oddball paradigm test to scale olfactory functionality instead. Then, to avoid disruption from surgery, a separate mouse cohort was used to examine the neurological consequences of the olfactory system in repetitive TBI mice. Because an increased phosphorylated tau (p-Tau) and tauopathy is known to be a major pathological phenotype in TBI, we examined the change of p-Tau level and its correlation with neuronal damage in the olfactory regions by TBI [20]. Using an interdisciplinary approach, we found alterations in olfaction-related EEG activities and found pathophysiological signs of neuronal atrophy, such as an increase of p-Tau signal and cell shrinkage in olfactory related areas.

## 2. Materials and Methods

### 2.1. Mice

A total of 47 male mice were used for this study, and detailed information is in Appendix A. Thy1(B6.Cg-Tg (Thy1-COP4/EYFP)18Gfng/J, The Jackson Laboratory, Bar Harbor, ME, USA) and wild-type mice (C57BL/6N; 12 weeks) were used. All TBI model mice received five times, spaced 3 days, of weight-drop-induced TBI events. While Thy1 mice were used to collect odor-evoked electrophysiology data, mice were used to collect immunochemistry data at three days after TBI, and mice were used to collect Y-maze and buried-food-seeking behavioral data. All mice were born and raised under standard laboratory conditions of a 12-h light/dark cycle at 22 °C, with 55% humidity, and were fed a standard laboratory diet and water ad libitum. The animal procedures were approved by the Institutional Animal Care and Use Committee (IACUC) of Korea Institute of Science & Technology (KIST).

### 2.2. TBI Model

Mice received a total of five weight-drop-induced TBIs, each administered every three days (Figure 1). Closed diffuse TBIs were performed by a weight-drop device (weight 100 g, fall height 75 cm, angle 90 deg), as previously described, but with a slight modification [21,22]. Control and TBI mice were anesthetized with 2% avertine (23 μg/g) IP injection before receiving the weight-drop-induced TBI. The head of mice was not fixed, and the impact was loaded at the prone position. The anatomical locus of impact was adjusted to bregma −1 to +4. Sham-injured animals were subjected to the same protocol, but no mass was dropped (Figure 1). No mice were killed following impacts, and mice were monitored for hemorrhage or skull fracture after TBI through observation with the naked eye. The severity of this model belongs to a mild TBI [21,22].

### 2.3. Immunohistochemistry

Immunohistochemistry was performed as previously described [22]. Three days after the last weight-drop-induced TBI, animals were deeply anesthetized (2% avertine (23 μg/g) IP injection) and then perfused through the ascending aorta with 100 mL of cold 0.1 M phosphate buffer saline (PBS), followed by 100 mL of 4% paraformaldehyde in 0.1 M PBS. The brains were removed, post-fixed overnight, and cryoprotected in 30% sucrose in 0.1 M PBS at 4 °C. Then sections were cut by cryotome into sections of 30 μm in thickness and processed by immunostaining. The sections were quenched endogenous peroxidase by 0.3% H_2_O_2_ and blocked with blocking solutions (5% normal goat serum in TBST (0.3% triton-X100 in TBS)), at room temperature (RT), for 2 h. Then the sections were incubated total tau (1:200; TAU-5; Abcam, Waltham, MA, USA) and AT8 (1:200; p-Tau (S202/T205); Thermo Fisher Scientific, Waltham, MA, USA) in blocking solution for 24 h. After three times of washing, the sections were incubated with biotinylated secondary antibody (1:200, Abcam, USA) in blocking solution at RT for 2 h and were conjugated with ABC solution (Vector Lab, Burlingame, CA, USA) at RT for 30 min. Immunoreactive signals were developed with DAB reaction solutions (Thermo Fisher Scientific, Waltham, MA, USA) at RT for 30~60 s and washed 3 times with tap water for 5 min. The nuclei of sections were counterstained with hematoxylin solution (Vector labs, Burlingame, CA, USA) for 1 min. After washing with tap water, the sections were dehydrated by serial concentration of EtOH and xylene. The sections were then permanently mounted, and the images were acquired by bright field microscopy. The images were analyzed by using Image J software with color deconvolution (Fiji, NIH, Bethesda, MD, USA). For immunofluorescence staining of APP, the sections were incubated with rabbit anti-amyloid protein precursor (APP) (1:200; Novus, Centennial, CO, USA) at 4 °C, overnight, and then incubated with goat anti-rebbit-488 (1:200; Abcam, Cambridge, MA, USA) at RT for 2 h. The images were acquired by immunofluorescence microscopy (Imager 2, Zeiss, German) and confocal microscopy (FlouView, Olympus, Tokyo, Japan). 

### 2.4. Cresyl Violet (CV) Staining

To examine neural size in the brain sections, cresyl violet (CV) staining was used in gelatin-coated microscopy slide by immersing the sections into distilled water at 1.0% (*w*/*v*) dissolved cresyl violet acetate (Sigma, Saint Louis, MO, USA) and glacial acetic acid (Sigma, Saint Louis, MO, USA). Then the sections were dehydrated by serial ethanol solution and mounted with toluene solution (Fisher Chemical, Waltham, MA, USA) [23].

### 2.5. Electrode Implantation

EEG electrodes were implanted after TBI treatment. Mice were deeply anesthetized with ketamine (120 mg/kg, intraperitoneal) and xylazine (6 mg/kg, intraperitoneal) and fixed in a stereotaxic apparatus (David Kopf Instruments, Model 902, Tujunga, CA, USA). Sterilized micro-screw electrodes (Asia Bolt, Incheon, Korea) were fixed onto the skull surface of the olfactory bulb (anteroposterior, 4.8 mm; mediolateral, 1.2 mm; dorsoventral, −1.1 mm from bregma), frontal (anteroposterior, 0.5 mm; mediolateral, 1.2 mm), and parietal cortex (anteroposterior, −3.08 mm; mediolateral, 3.75 mm), with ground/reference electrodes implanted on the intraparietal bone. The electrode coordinates were selected in accordance with the mouse atlas. To secure the electrode positions, dental cement (VertexTM Self-Curing, Vertex-Dental, Soesterberg, The Netherlands) was applied, along with two polycarbonate nuts (inner diameter of 3 mm, Nippon Chemi-Con, Japan), which were attached to the caudal edge of the cement for head-fixation during the experiment.

### 2.6. LFP (Local Field Potential) Recording

After one week of recovery period, mice were placed in front of the olfactometer while being head-fixed in a custom mouse restrainer. Mice were placed so that their nose tip had a 1 cm distance from the olfactometer outlet. Three-channel LFP (the olfactory bulb, the frontal cortex, and the parietal cortex) data were collected during the olfactory oddball paradigm with a Cerebus amplifier (Blackrock Microsystems, Salt Lake City, UT, USA). All signals were digitized with a 2 kHz sampling rate and bandpass filtered from 0.3 to 500 Hz.

### 2.7. Olfactory Oddball Paradigm

The experiment was conducted by utilizing the olfactometer setup previously described by Kum et al. (2019) [20] (Figure 1D). For odor delivery, constant flow of filtered air (1 L/min) was delivered, with odor stimuli being diluted by pumped air (200 mL/min). Methyl salicylate (Sigma Aldrich, St. Louis, MO, USA, >99% purity, mineral oil solution, odorant:solvent ratio was 3:1) and ethyl acetate (Sigma Aldrich, MO, USA, >99.5% purity, distilled water solution, odorant:solvent ratio was 1:1) were selected for the olfactory oddball paradigm, with methyl salicylate being the standard odorant and ethyl acetate being the deviant odorant. The standard and deviant odor were presented randomly with a 5:1 ratio. The stimulation period was 2 s with an inter-stimulus interval of 20 s. After being presented, odor stimuli were vacated through the vacuum pump (1.5 L/min) (Figure 1D).

One session was composed of 90 trials (75 standard trials and 15 deviant trials), lasting approximately 33 min (Figure 1D). Each mouse received four sessions of experiment in total, which were divided into two days in order to prevent adaptation to the odor stimuli and over-exhaustion caused by the long duration of head-fixing. Two sessions were given each day, with an interval of 30 min between each session, and there were three to five days between the two days of experiment. The recording room was ventilated for 30 min after each recording session.

### 2.8. LFP Analysis

The LFP data were processed in Matlab 2019a version (Mathworks, Natick, MA, USA) with Signal Processing Toolbox (ver. 8.2). First, the signals were bandpass-filtered (1–150 Hz, using a 5-th order Butterworth filter, then transformed into time-frequency domain using sliding Hanning window (512 ms length, 100 ms resolution) and fast Fourier transform to obtain amplitude spectrogram. The frequency bands of interest were delta (1.5–4 Hz), theta (4–8 Hz), beta (12–30 Hz), low gamma (40–60 Hz), and high gamma (70–120 Hz) bands. In case of analysis of resting state, the amplitude spectrum of each channel was obtained from each individual by averaging the spectrogram of 2 min before the olfactory oddball paradigm (Figure 1D). 

To investigate the cross-frequency coupling across different frequency bands of oscillations, we calculated the modulation index (MI), as described by Tort et al. [24]. Briefly, the raw EEG epochs were bandpass filtered for the slow and fast frequency bands of interests, using Burtherworth filters, followed by Hilbert transform to extract the phase values of the slow rhythm and the magnitudes of the fast rhythm. Then, the averaged magnitudes of fast rhythm was obtained as a function of phase of slow rhythm and used as the probability distribution for Shannon entropy. MI was calculated by z-score of Shannon entropy with respect to the uniform distribution. In the current study, the bin size of phase was π/15.

For the analysis of olfactory oddball paradigm, event-related spectral perturbation (ERSP) was obtained by averaging the spectrogram of baseline-corrected signals with temporal and frequency resolutions of 100 ms and 2 Hz. To match the number of trials between standard and deviant trials, only the standard trials right before the deviant trials were used. To evaluate the diminished oscillatory activities in TBI model mice, independent sample *t*-tests (alpha = 0.05, one-tailed) were performed by using ttest2.m function in MATLAB 2019a, over the individual mean value in the band-of-interests and time-of-interests (degree of freedom = 15).

### 2.9. Olfactory Function-Associated Behavior Tests 

#### 2.9.1. Y-Maze Test

Y maze was performed the subsequent day after the fifth weight drop at 12 weeks postnatal days, as previously described with a slight modification of procedures [9]. The Y maze apparatus was composed of three enclosed arms (35.3 cm (length) × 6 cm (width) × 18.4 cm (height)) constructed of white acrylic. A tray of home bedding (7.2 cm (length) × 2.9 cm (width) × 1.3 cm (height)) was placed at the end of one arm, and the same to its adjacent arm with new bedding. Each trial began by placing each individual mouse in the arm with no bedding and was observed for 10 min. Y maze apparatus was cleaned by 70% ethanol and distilled water every trial. Total distance (cm), velocity (cm/s), time (s), frequency (count), and latency to start (time) at each arm were recorded by Ethovision Software (Noldus Information Technology, Wageningen, The Netherlands).

#### 2.9.2. Buried-Food-Seeking Test

To identify the olfactory function, the buried-food-seeking test was used as described previously [25,26]. Before the test, mice were deprived of food for 24 h, but they received water in the home cage. In the test under the video recording system (1.0 Megapixel USB camera, ELP, China), mice were in the clean cage of regular size (25.5 cm (length) × 20 cm (width) × 13 cm (height)), with a 5 cm layer of cage bedding (Hygiene bedding, SAFE, Germany). Then 1~2 g of a pellet was buried at a depth of 2 cm in the surface of the fresh bedding and was changed every test with bedding after cleaning the cage. Mice were in the cage for 10 min with video recordings, and the behavior data were analyzed by video tracking software (EthoVision XT version 13, Noldus, The Netherland). In this study, we determined latency to first finding the hidden food pellet, duration, distance moved, and velocity in buried food area (10 cm × 8 cm). The heat maps were presented with the average of distance moved. 

### 2.10. Statistical Analysis

For testing EEG responses, non-parametric versions of ANOVA (Kruskal–Wallis test) were performed (alpha = 0.05, two-tailed), using *kruskalwallis* in Matlab2017b (Mathworks, Inc., Natick, MA, USA). For continuous data with a normal distribution, an unpaired *t*-test (two-tailed) was used to examine significant differences between two groups.

## 3. Results

### 3.1. Repetitive TBI Elevates Amyloid Precursor Protein (APP) Immunoreactivity in the Olfactory-Bulb-Associated Areas

In order to examine the effects of repetitive TBI on the olfactory function and the olfactory-bulb-associated neuropathology [10,11,25], we applied our established weight-drop-induced TBI model [21,22], which was administered five times for two weeks, with an interval of three days (Figure 1). At first, we performed immunohistochemistry and detected the APP level, a marker of axonal damage, in the olfactory track and bulb-associated regions (Figure 2). The APP immunoreactivity was markedly increased in the lateral olfactory (LO) tract (Figure 2). The densitometry analysis showed significant increases of APP level in the lateral olfactory tract (LO), the medial part of anterior olfactory nucleus (AOM), and the prelimbic and orbital cortex (Figure 2).

### 3.2. p-Tau Level and Neuronal Cell Size Are Inversely Correlated in Repetitive TBI

It has been reported that mild-TBI patients have acute axonal accumulations of total and phospho-Tau (p-Tau) within hours to weeks [26,27]. The densitometry analysis showed a significant increase in the p-Tau (Ser202/Thr205) level in the olfactory-bulb-associated region, including the lateral AOL, medial AOM, granular cell layer of the olfactory bulb (GrO), tenia tecta (dorsal DTT), main olfactory bulb granular cell layer (MOBgr), olfactory tubercle (OT), pyriform cortex, and entorhinal cortex (EC) (Figure 3A–C). Previous studies have shown that an increase of total Tau and p-Tau levels are a pathological marker in chronic traumatic encephalopathy (CTE) and Alzheimer’s disease (AD) patients and TBI animal models [28,29]. Accordingly, we also checked the total Tau level in the olfactory-bulb-associated region of TBI and control mice (Figure 4). The total Tau levels were not significantly changed in the various olfactory-bulb-associated areas, such as the AOL, AOM, GrO, DTT, MOBgr, and pyriform cortex between TBI and control mice (Figure 4C). These data exhibit that the increase of p-Tau levels is specifically associated with pathological changes in the olfactory-bulb-associated region of TBI mice.

To better understand whether the p-Tau level is associated with neuronal atrophy or neuronal cell size shrinkage in the olfactory-bulb-associated areas, we performed cresyl violet (CV) staining combined with cell size analysis on the regions displaying the elevated p-Tau immunoreactivity (Figure 3 and Figure 5). When neuronal cell size was verified, the AOL, AOM, GrO, and DTT areas exhibited significant reduction of the neuronal size (control; *N* = 5 and repetitive TBI; *N* = 5) (Figure 5A,B). Additionally, we ran correlation analysis between p-Tau and neuronal cell size in the olfactory-bulb-associated areas (Figure 5C). The linear regression analysis revealed that neuronal cell size was inversely correlated with the p-Tau level, AOL, GrO, and DTT (Figure 5C). These data implicate that the pathological increase of p-Tau levels can induce neuronal damage in the olfactory-bulb-associated area in TBI mice.

### 3.3. Repetitive TBI Alters Spectral Power of Slow Oscillations of Olfactory Bulb

We investigated the effects of repetitive TBI on spontaneous oscillation by measuring the LFP in the prefrontal cortex, hippocampus, and olfactory bulb. The level of spontaneous oscillations was quantified with power spectral analysis and is summarized in Table 1. Compared to the control group (*N* = 4), mice in the repetitive TBI group (*N* = 10) exhibited a significant decrease in delta bands of the olfactory bulb, whereas spontaneous oscillations in the prefrontal cortex and hippocampus were not affected in the repetitive TBI group (Figure 6A,B).

We further investigated the cross-frequency coupling by calculating the modulation index (MI) according to Tort, Komorowski, Eichenbaum, and Kopell (2010) [24]. The cross-frequency coupling is an interaction between slow and fast brain rhythms, appearing as modulation of fast rhythm by phase of slow rhythm, and has been known as a key mechanism of entraining large-scale distributed neuronal networks required for cortical processing at behavioral timescales [30]. We calculated the MI for delta and theta phases. We found that MI values for the delta phase were significantly attenuated in fast oscillations of the olfactory bulb of repetitive TBI mice. The comodulograms, heat maps showing MI in frequency and phase domains (Figure 6C), display the averaged MI across animals for the control mice and for the repetitive TBI mice, with differences largely captured by changes in scale. Statistical tests returned that the cross-frequency couplings between delta phase and beta/low gamma amplitude are significantly reduced in the repetitive TBI model (Figure 6D).

### 3.4. Repetitive TBI Attenuates Response to Deviants

We performed an olfactory oddball paradigm with methyl salicylate as a standard odor and ethyl acetate as a deviant odor. Figure 6A shows the averaged ERSP for standard and deviant stimuli. Overall, odor elicited the brain rhythms in all frequency bands, and the response of the beta power was the largest among them. In addition, the oddball odor stimuli induced significantly stronger responses than the standard odor stimuli. Figure 6B shows the grand average of differential spectrograms of ERSP calculated from individual mice, showing an overall debilitation. Figure 7C,D marks the ranges of frequency, showing the significant or marginal differences in responses to the deviant odor in the repetitive TBI mice (*N* = 10) compared to the control mice (*N* = 4). The responses in the measured areas are summarized in Table 2. It is noteworthy that the relative power of dominant power (beta) dropped from 65% (control) to 35% (TBI), but in a marginally significant way. On the other hand, the other frequency EEG bands were enhanced in a significant way by repetitive head injury, as summarized in Table 2.

### 3.5. Repetitive TBI Mice Exhibit Decline of the Olfactory Function-Associated Behaviors

We used a Y maze apparatus to characterize the effects of repetitive TBI on olfactory-related behaviors (Appendix A, Figure 8A and Appendix A). Although both control (*N* = 5) and repetitive TBI mice (*N* = 6) displayed a preference for home bedding (Figure 8B), repetitive TBI mice exhibited longer duration in new bedding versus control mice. Moreover, we found that the amount of time or duration spent in the closed arm with home bedding was slightly different between groups, but there was no statistical significance (Figure 8C). To further verify whether TBI affects the olfactory-function-associated behavior, we further performed a buried-food-seeking test (Figure 8D). As we expected, the latency to first finding the hidden food pellet, a major behavioral parameter, was significantly delayed in repetitive TBI mice compared to control group of mice (Figure 8F). In contrast, the duration and distance moved in buried food area were significantly elevated in repetitive TBI mice (Figure 8F). Interestingly, repetitive TBI mice showed an increase of velocity in the area of the hidden food pellet (Figure 8F). These data suggest that reductions of olfactory-function-associated behavioral tasks are correlated with pathological and physiological changes in the olfactory-bulb-associated area of TBI mice.

## 4. Discussion

Olfactory dysfunction is an indicator of dementia risk; however, whether it is present in individuals with a history of repetitive TBI has yet to be fully elucidated. As it is not easy for conventional neuroimaging to reveal the underlying symptoms and abnormalities [31], we have demonstrated the structural and functional changes in the olfactory system by using a repetitive TBI mouse model. Particularly, repetitive TBI animals showed evident pathology in the olfactory-bulb-associated areas at three days after the injury. Interestingly, the olfactory bulb, a region anatomically larger and possibly more vulnerable to injury in mice, was not prone to pathology. Similar pathological sequelae have been reported in other repetitive TBI models showing damage in rostral areas (optic tract, interior olfactory nuclei, and lateral olfactory tract) after three repetitive TBI events [13]. Rather, it appears that regions directly caudal, adjoining the olfactory bulb, may be more vulnerable to cellular damage.

It is known that olfactory dysfunctions of TBI patients significantly decrease quality of life (QoL) score [32]. Moreover, olfactory dysfunctions are observed in the early phase (24 h) of mild TBI patients who have post-concussion and anxious symptoms at long-term phase (1 year) [33]. Our current data suggest that mild repetitive TBI mice show impairments of the olfactory-related behavioral symptoms (Figure 8), and this TBI mouse model is a good tool to find the mechanisms of olfactory dysfunctions in both acute and long-term phase of mild TBI patients. The TBI model can be classified into sport-related acute TBI models and mild repetitive TBI models [34,35]. Mild repetitive TBIs are founded in boxers, football players, and wrestlers, those who may have axonal and cytoskeletal alterations and tauopathy in the brain [36]. Indeed, these types of TBI patients and animals models exhibited a strong diffuse axonal injury, and the patients have symptoms of loss of consciousness, anxiety, fatigue, headache, and concussive convulsion and/or impact seizure [37]. Mild TBI patients also have olfactory dysfunctions, such as anosmia, the total loss of the sense of smell [35,38]. Our repetitive TBI mice showed a significant change of olfactory dysfunction-related behaviors compared to control mice. Both the axonal damage in the olfactory-bulb-associated areas and the loss of olfactory-function-associated behaviors in our TBI animal model provide the clinical relevance of olfactory dysfunction in human TBI patients. Thus, this TBI animal model can be useful for further validating a precise mechanism of the olfactory dysfunctions of mild-TBI patients [34].

It has shown that mild concussive TBI patients exhibit acute axonal accumulations of p-Tau within hours to weeks [26,27]. TBI-induced tauopathy in mice and humans is observed in both axonal and subcellular compartments through various molecular mechanisms, such as dysregulation of kinases and protein phosphatases [39]. Furthermore, a recent clinical study has demonstrated that plasma p-Tau levels and p-Tau/total Tau (T-tau) ratio in acute and chronic TBI patients are the better diagnostic and prognostic biomarkers than only T-tau level [28]. These phenomena are also observed in the mild repetitive TBI mouse model [40]. Taken together, our study suggests that the p-Tau level and p-Tau/T-Tau ratio are also neuropathological biomarkers for the olfactory-bulb-associated regions in the mild repetitive TBI mouse model.

Studies have shown that repetitive TBI events lead to neuropathological changes. For example, Mouzon et al. observed persistent white matter loss and corpus callosum thinning up till 12 months after repetitive TBI injury [15]. Furthermore, the same study demonstrated that any history, even a single TBI event, may be significantly injurious and could be correlated with cognitive decline or neurobehavioral differences [15]. In the countercoup brain-injury model, Small et al. reported that TBI induces the dysregulation of glutamate dysfunction in reactive astrocyte, resulting in an increased susceptibility to kainic acid–induced seizures [41]. In this regard, cell shrinkage, a characteristic of programmed cell death or apoptosis, may be an indicator of long-term cognitive impairments and reduced olfactory capacity. Whether the pathological sequelae in the olfactory system will persist and evolve post TBI in the olfactory system remains to be clarified.

In this study, repetitive TBI produced subtle deficits in olfactory performance, as measured in the y-maze task equivalent activity levels, and the velocity between control and TBI mice showed a loss of preference for home bedding in TBI mice. We propose that it was not due to sensorimotor disturbances, but, in fact, it was due to a loss in olfactory discrimination (Appendix A). We performed the buried-food-seeking test to evaluate olfactory function in the TBI models (Figure 8). It also showed significant changes in latency and duration for seeking buried food in TBI mice. However, we can also use other behavioral tests, such as olfactory-cued fear conditioning, which detects freezing and fear-potentiated startle associated with the odor cue and electrical foot shocks [42]. It is a good animal model for examining the olfactory region and amygdala circuits effectively [43]. We are looking forward to use this behavior model for the future study.

The subtle differences in olfactory performance between TBI and control mice are in alignment with the effects of repeated TBI in humans. With a recent study that examined the olfactory functionality of former National Football League (NFL) players, former NFL players who experienced repetitive head impacts showed a lower performance in the Brief Smell Identification Test (B-SIT) compared to healthy controls. However, the majority of subjects (~94%) performed within healthy olfactory standards [11]. Although football players may experience, on average, more than 1000 subconcussive events per playing season, olfactory performance remains to be a subtle factor in describing brain injury. Similar results were obtained in analyses measuring the visuospatial scanning speed, visuomotor processing speed, visuospatial construction, and verbal memory of NFL players compared to healthy controls [44]. Subtle olfactory differences may be difficult to detect with the current behavioral assessments used on human TBI patients; thus, this warrants further attention.

To overcome this weakness in behavioral assessments of olfactory functionality, here we used EEG recordings of repetitive TBI mice to detect olfactory dysfunction in repetitive TBI patients. Known also as respiratory rhythms, olfactory slow oscillations, especially the delta band oscillation, are commonly observed in the mouse olfactory bulb, primary olfactory cortices, frontal cortex, and hippocampus, and are synched to the respiratory cycle [45,46]. Even though there are not many previous studies concerning the olfactory delta oscillation, a recent study proposed that respiration-coupled slow oscillation activities play a major role in transferring information between olfactory-related brain regions to high-order brain regions [47]. This suggests that weakened resting-state delta oscillation at the olfactory bulb of TBI mouse models could reflect alterations in internal processing, thus causing impaired olfactory perception.

Massive interconnected parallel processes in the brain require the coordination of information between areas, and this often appears as cross-frequency couplings (CFCs) [48,49]. The reduction in resting state CFC in TBI mouse is in line with previous studies that found in a reduction of functional connectivity strength within the delta-beta and delta-gamma1 frequency pairs in the frontal brain region of TBI human patients [50]. Recent studies have also been reporting the modulation of high-frequency oscillations, such as local beta rhythm and gamma rhythms, by slow rhythms in the OB [34] and other higher-order brain regions [51,52], suggesting that the impaired resting state delta power could be the cause of the decreased CFC between delta and beta/low gamma band frequencies in repetitive TBI mouse models.

The present study also used the olfactory oddball paradigm to measure direct brain responses to odor stimuli. Oddball paradigms are frequently used to study sensory discrimination by comparing event-related potentials (ERPs) or event-related spectral power (ERSP) from a standard stimulus with high probability to a deviant (oddball) stimulus with low probability. Typically, oddball stimuli elicit stronger responses in a correlational way of discriminability; thus, this paradigm has been widely used in conscious patients, infants, or animals that have limited expression in their language or motion. Here, using the olfactory oddball paradigm, we measured LFPs in the olfactory bulb (OB), which serves as the direct receiver of olfactory sensory signals in the mouse brain, and other higher-order regions, such as the prefrontal cortex and hippocampus.

Our data showed significant neural impairments in the OB, whereas the prefrontal cortex and hippocampus appeared to be unaffected by the injury. This is in line with past studies that show weakened brain activities in olfaction-related brain regions in PD patients who showed reduced olfactory functioning [53]. The power of beta oscillations was especially observed to be decreased in the OB. The beta oscillation is known to be one of the main oscillatory rhythms linked to odor processing, commonly observed at the OB. The beta oscillation is reported to be strongly elicited by aversive odors with high volatility [25,54]. Moreover, past studies have shown enhanced beta power induced by repeated exposure to the same series of odorants [55], suggesting that the beta oscillation is closely linked to odor recognition and classification. Thus, decreased beta power in the OB during olfactory oddball tasks shows impaired functionality in these specific areas of odor perception.

## 5. Conclusions

In conclusion, we showed that, after repetitive TBI events, olfactory-related areas demonstrate abnormal electrophysiological changes, as well as increased p-Tau signal and decreased cell size. Tau-phosphorylation causes cellular dysfunction, resulting in decreased cell size associated with altered pathophysiology in olfactory-related areas. These findings demonstrate that a history of repetitive TBI increases pathophysiology, which may serve as a criterion of post-TBI neurodegenerative diseases, such as chronic traumatic encephalography and dementia [56]. Our work further suggests that electrophysiological neural data can aid in revealing the olfactory impairment frequently observed in repetitive TBI, and the possible progression of brain damage in patients. Given the subtle behavioral changes observed during odor-perception tests, patients with a history of repetitive TBI may be unaware of their condition. Without the use of verbal or behavioral examinations, the early diagnosis of anosmia by its related EEG responses may be a more objective method for measuring brain damage.

## Figures and Tables

**Figure 1 biomedicines-10-00865-f001:**
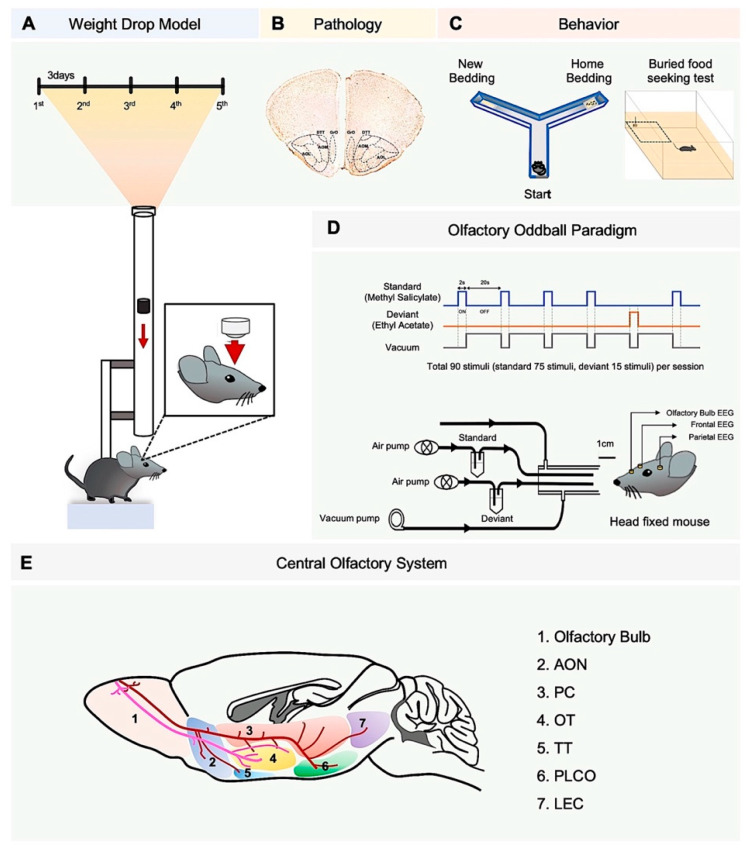
Weight-drop model of traumatic brain injury (TBI) and experiment workflows. (**A**) Mice experienced 5 weight-drop-induced TBI events. (**B**) Brain tissue was analyzed from postmortem WT mice (*N* = 5 Control, *N* = 5 TBI) within 5 days after 5th TBI. (**C**) Living WT mice underwent Y maze behavioral test (*N* = 5 Control, *N* = 6 TBI) and buried-food-seeking test (*N* = 6 Control, *N* = 6 TBI). (**D**) A simplified diagram of the olfactometer used to deliver the two odor stimuli. Schematic overview of the two-odor olfactory oddball paradigm (**upper panel**). Odor-evoked electrophysiology data of Thy1 mice was collected by using a two-odor olfactory oddball paradigm (*N* = 4 Control, *N* = 10 TBI) (**lower panel**). (**E**) A schematic diagram of the central olfactory system. Axonal projections from the olfactory bulb reach a number of areas, including the anterior olfactory nucleus (AON), pyriform cortex (PC), olfactory tubercle (OT), tenia tecta (TT), lateral entorhinal cortex (LEC), and cortical amygdala (PLCo).

**Figure 2 biomedicines-10-00865-f002:**
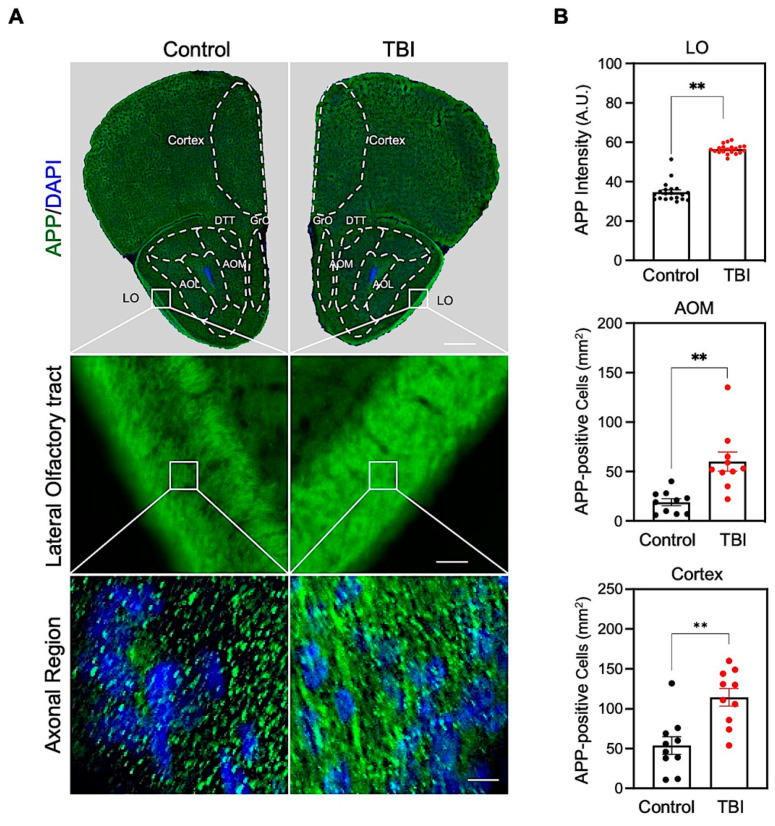
Olfactory-bulb-associated areas exhibit increased amyloid protein precursor (APP) level in TBI mice. (**A**) Representative hemisphere images of the olfactory bulb region (**top panel**), lateral olfactory (LO) tract region (**middle panel**), and high magnification of LO regions in control and TBI mice (bottom panel). Scale bars (white): 500 μm (**top**), 200 μm (**middle**), and 10 μm (**bottom**). (**B**) Densitometry analysis showed a significant increase of APP level and APP-positive cell number in repetitive TBI mice compared to control mice (control, *N* = 5; TBI, *N* = 5). Error bars indicate SEM; **, significantly different from control at *p* < 0.01.

**Figure 3 biomedicines-10-00865-f003:**
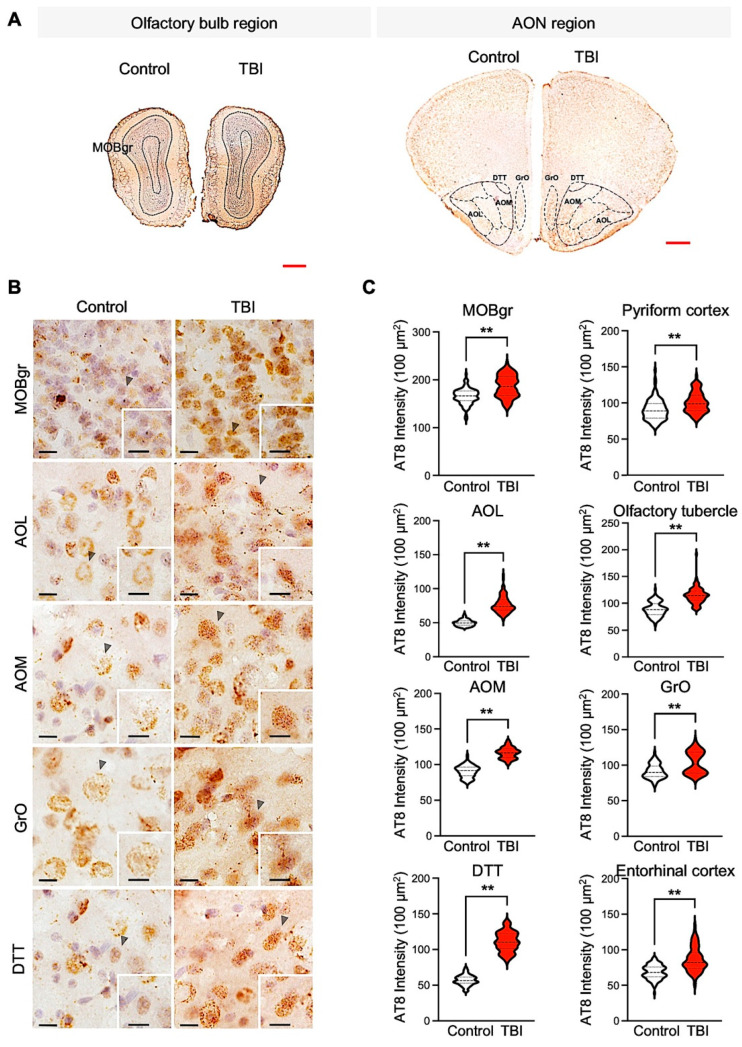
Olfactory-bulb-associated areas exhibit increased phosphorylated tau (p-Tau) level in TBI mice. (**A**) Representative hemisphere images of the olfactory bulb region (**left panel**) and AON region (**right panel**) in control and TBI mice. Scale bars (red): 500 μm. (**B**) TBI increased the immunoreactivity of p-Tau (Ser202/Thr205) in the olfactory bulb region. Scale bars (black): 10 μm. (**C**) Densitometry analysis showed a significant increase of p-Tau level in repetitive TBI mice compared to control mice (control, *N* = 5; TBI, *N* = 5). Error bars indicate SEM; **, significantly different from control at *p* < 0.01.

**Figure 4 biomedicines-10-00865-f004:**
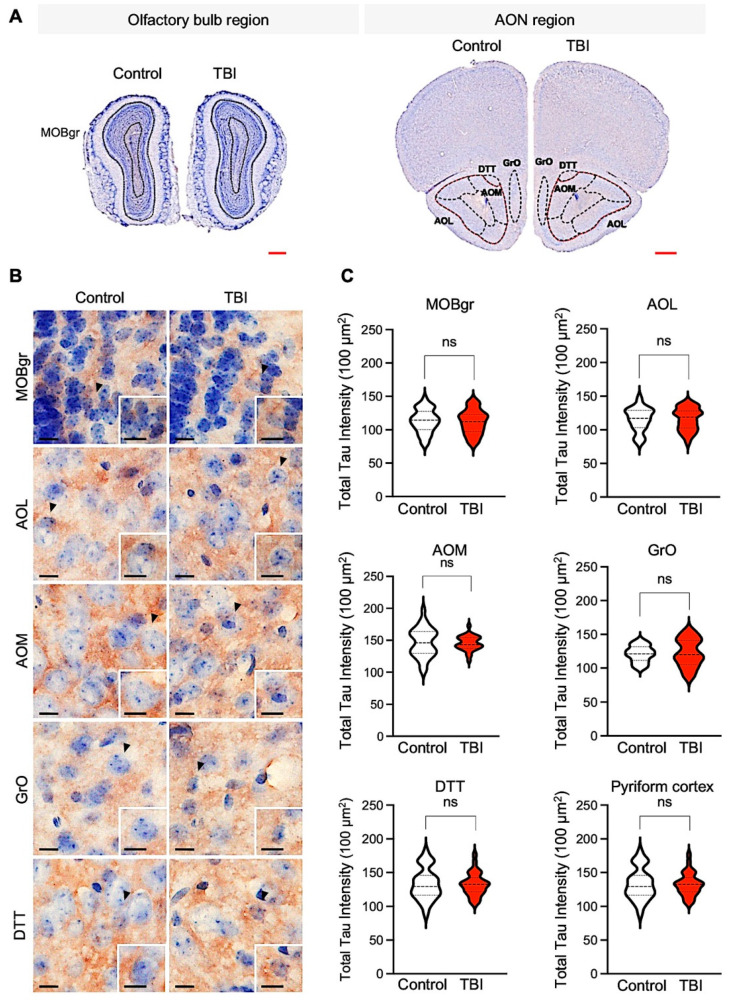
Olfactory-bulb-associated areas exhibit no difference of total tau level in TBI mice. (**A**) Representative hemisphere images of the olfactory bulb region (coronal section) in control and TBI mice (**upper panel**). Scale bars (red): 500 μm. (**B**) High-magnification images of the olfactory bulb region in control and TBI mice. Scale bars (black): 10 μm (**C**) Densitometry analysis showed no difference in total tau level in repetitive TBI mice compared to control mice (control, *N* = 5; repetitive TBI, *N* = 5). Error bars indicate SEM; ns, not significant.

**Figure 5 biomedicines-10-00865-f005:**
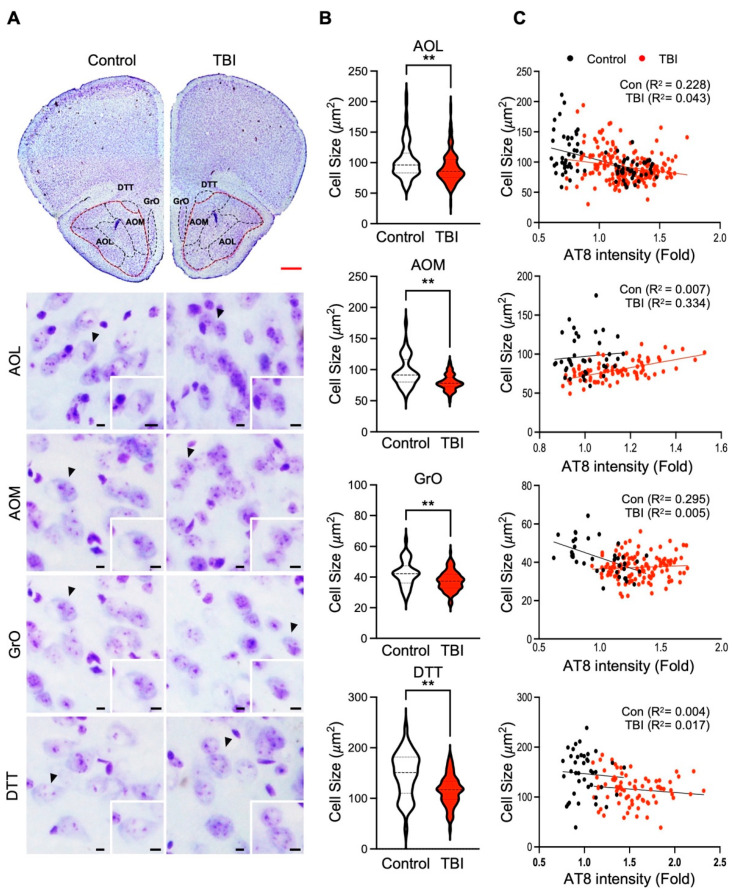
Neuronal size in olfactory-bulb-associated areas is inversely correlated with p-Tau by repetitive TBI. (**A**) Neuronal size was decreased in repetitive TBI mice compared to controls. The brain tissues were stained with cresyl violet (CV). Scale bars: 500 μm (red) and 10 μm (black). (**B**) Cell size analysis showed a significant decrease in repetitive TBI compared to control mice in the AOL, AOM, GrO, and DTT. (**C**) Linear aggression analysis exhibited an inverse correlation between p-Tau level and neuronal size in AOL, GrO, and DTT regions (control, *N* = 5; TBI, *N* = 5). Error bars indicate SEM. Significantly different from control at ** *p* < 0.01.

**Figure 6 biomedicines-10-00865-f006:**
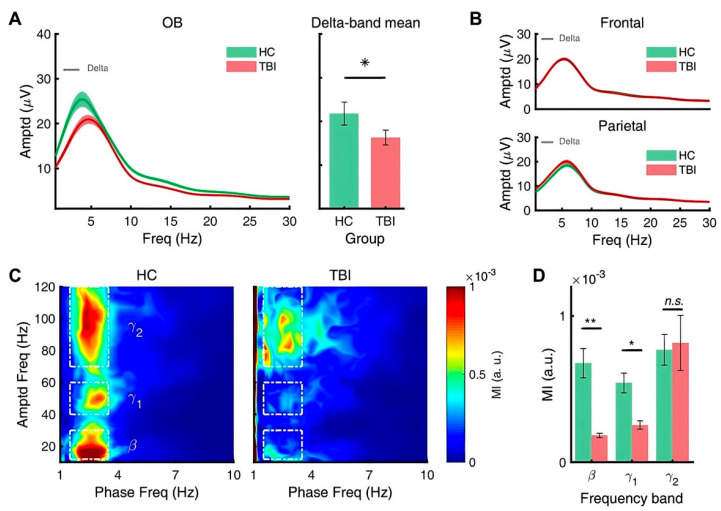
Olfactory bulb exhibits reduced delta oscillations and modulation index by repetitive TBI. (**A**) Resting state phase amplitude analysis in the olfactory bulb. There was a decrease (*p* = 0.0309) in resting state delta-band amplitude in TBI mouse models. (**B**) There was no significant amplitude change observed at the frontal and parietal regions of the mouse brain. (**C**) Comodulogram comparing healthy and TBI mouse subjects’ cross-frequency coupling with delta bands. TBI mouse models showed reduced cross-frequency coupling in delta–beta and delta–theta1 phase pairs. (**D**) Statistical analysis illustrating lowered MI of delta–beta (*p* = 0.0077) and delta–theta (*p* = 0.0289) oscillation pairs of TBI mouse models. Error bars indicate SEM. Significantly different from control at * *p* < 0.05 and ** *p* < 0.01.

**Figure 7 biomedicines-10-00865-f007:**
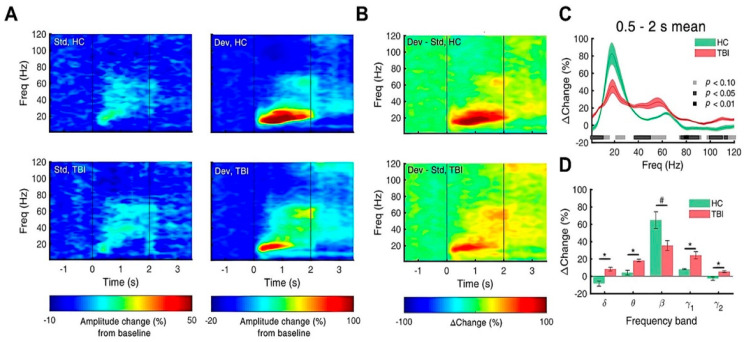
Response to deviant odors is attenuated by repetitive TBI mice. (**A**) Power spectrogram illustrating averaged ERSP for standard and deviant odors. Deviant odors elicit a stronger ERSP response, especially in beta band range. (**B**) Differential spectrogram of ERSP response. (**C**) Mean of amplitude difference during 0.5–2 s time range. (**D**) Statistical analysis of amplitude difference of each band range. Relative power of beta oscillation decreased (*p* = 0.0985), while other frequency ranges showed increased relative power (*p* < 0.05). Error bars indicate SEM. Significantly different from control at * *p* < 0.05 and no significance indicated as #.

**Figure 8 biomedicines-10-00865-f008:**
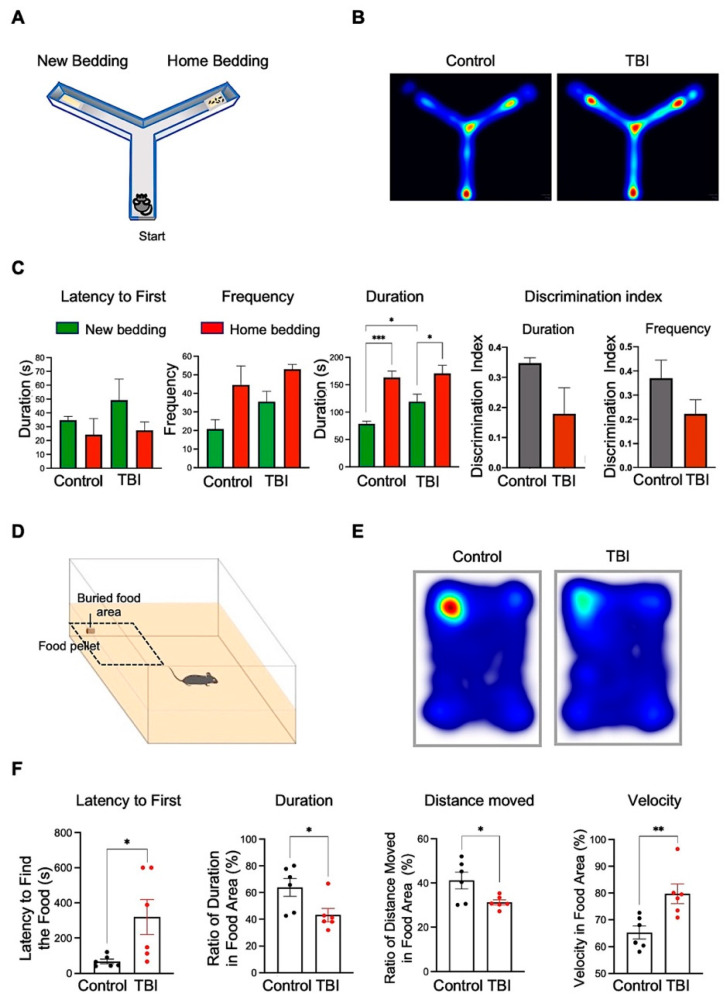
Repetitive TBI reduces olfactory-function-associated behaviors in home bedding and buried-food-seeking tests. (**A**) A scheme for Y maze apparatus and placement of new bedding and home bedding. (**B**) A representative heat map of Y maze trials shows an increased frequency in TBI mice in the arm with new bedding. (**C**) Control and TBI mice display alterations of latency to first, frequency, and duration. TBI mice show greater frequency for home bedding (*p* = 0.0189), as well as a greater duration for new bedding (*p* = 0.0276), compared to controls. Discrimination Index for duration and frequency shows that control mice have a greater preference for home bedding (*p* = 0.1154, *p* = 0.1512) (control, *N* = 5; TBI, *N* = 6). (**D**) A scheme for buried-food-seeking test apparatus and placement of a mouse. (**E**) A representative heat map of buried-food-seeking trials show an increased latency to first in TBI mice. (**F**) Control and TBI mice display alterations of latency to first, duration, distance moved, and velocity. TBI mice show greater latency to first (*p* = 0.031), but decreased duration (*p* = 0.0276) and distance moved (*p* = 0.031), compared to controls (control, *N* = 6; TBI, *N* = 6). Error bars indicate SEM. Significantly different from control at * *p* < 0.05, ** *p* < 0.01, and *** *p* < 0.001.

**Table 1 biomedicines-10-00865-t001:** Summary of spontaneous oscillation and modulation by delta phase.

EEG Parameters	Frequency Bands	Frontal Cortex				Parietal Cortex				Olfactory Bulb			
		Control (*N* = 4)	TBI (*N* = 10)	*t-Stat* *(df = 15)*	*p*-Value	Control (*N* = 4)	TBI (*N* = 10)	*t-Stat* *(df = 15)*	*p*-Value	Control (*N* = 4)	TBI (*N* = 10)	*t-Stat* *(df = 15)*	*p*-Value
Amplitude (μV)	Delta	15.21 (1.99)	14.66 (2.33)	0.52	0.3056	12.99 (2.73)	13.85 (2.23)	−0.72	0.2425	23.30 (6.79)	17.52 (4.99)	2.02	**0.0309**
	Theta	16.40 (2.20)	15.81 (2.04)	0.57	0.2898	15.49 (3.05)	16.05 (3.85)	−0.33	0.3718	17.91 (3.46)	15.77 (3.17)	1.33	0.1013
	Beta	5.11 (0.80)	4.70 (0.51)	1.28	0.1100	4.80 (0.74)	4.77 (0.80)	0.07	0.4708	5.12 (1.04)	4.37 (0.37)	2.05	**0.0293**
	Low Gamma	2.59 (0.44)	2.39 (0.24)	1.17	0.1311	2.50 (0.44)	2.47 (0.24)	0.21	0.4191	2.88 (0.56)	2.73 (0.35)	0.69	0.2492
	High Gamma	1.52 (0.36)	1.48 (0.23)	0.26	0.4004	1.44 (0.39)	1.48 (0.19)	−0.27	0.3957	1.99 (0.44)	1.97 (0.29)	0.11	0.4586
Coupling to delta phase (×10^4^)	Beta	0.54 (0.35)	0.96 (0.54)	−1.89	**0.0388**	1.50 (0.67)	1.71 (0.62)	−0.69	0.2500	5.68 (4.32)	1.71 (0.65)	2.73	**0.0077**
	Low Gamma	0.49 (0.38)	0.86 (0.43)	−1.88	**0.0396**	0.89 (0.74)	1.10 (0.35)	−0.76	0.2284	4.73 (2.83)	2.53 (1.42)	2.05	**0.0289**
	High Gamma	0.72 (0.53)	1.23 (0.55)	−1.94	**0.0360**	0.61 (0.46)	1.36 (0.86)	−2.18	**0.0227**	7.22 (6.18)	7.11 (8.86)	0.03	0.4884

Results of analysis of resting state EEG. Independent *t*-test was used. The values are average and standard deviation in parenthesis.

**Table 2 biomedicines-10-00865-t002:** Summary of event spectral density during olfactory oddball paradigm.

EEG Parameters	Frequency Bands	Frontal Cortex				Parietal Cortex				Olfactory Bulb			
		Control (*N* = 4)	TBI (*N* = 10)	*t-Stat* *(df = 15)*	*p*-Value	Control (*N* = 4)	TBI (*N* = 10)	*t-Stat* *(df = 15)*	*p*-Value	Control (*N* = 4)	TBI (*N* = 10)	*t-Stat* *(df = 15)*	*p*-Value
Amplitude Difference (%)	Delta	−1.62 (9.99)	1.60 (15.87)	−0.49	0.3146	−15.94 (3.90)	−10.13 (11.15)	−1.40	0.0915	−7.38 (17.07)	9.49 (12.22)	−2.36	**0.0160**
	Theta	−4.26 (4.96)	2.70 (8.17)	−2.09	**0.0272**	−8.55 (2.41)	−1.48 (7.39)	−2.58	**0.0105**	7.69 (13.26)	20.43 (6.56)	−2.56	**0.0109**
	Beta	2.95 (7.65)	7.75 (12.05)	−0.96	0.1751	3.04 (11.60)	8.70 (13.43)	−0.92	0.1851	64.85 (54.11)	35.49 (34.52)	1.34	0.0985
	Low Gamma	6.10 (3.06)	10.49 (3.87)	−2.57	**0.0106**	6.63 (4.12)	11.09 (5.75)	−1.81	**0.0448**	8.19 (3.54)	24.44 (24.92)	−1.82	**0.0443**
	High Gamma	13.10 (8.50)	15.81 (10.49)	−0.58	0.2855	7.05 (6.64)	14.33 (7.68)	−2.08	**0.0277**	−3.43 (11.23)	4.84 (7.29)	−1.82	**0.0441**

The ratio (P_deviant_–P_standard_)/P_standard_ was calculated for each frequency band and then noted in percent, where P_standard_ and P_deviant_ are powers during standard and deviant odor stimulations, respectively.

## Data Availability

The datasets generated and analyzed during the current study are available from the corresponding author upon reasonable request.

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
