# Peer review of "Increases of Phosphorylated Tau (Ser202/Thr205) in the Olfactory Regions Are Associated with Impaired EEG and Olfactory Behavior in Traumatic Brain Injury Mice"

_biomedicines, 2022, doi:10.3390/biomedicines10040865_

Round 1

Reviewer 1 Report

The current study deals with damage to olfactory system caused by TBI in a mouse model. Mice were subjected to repetitive traumas to the head, histopathological examinations, electrophysiological recordings and behavioural function tests of olfaction were performed sequentially on TBI mice. A wide range of abnormality was observed in the multi-modal tests of olfactory system post TBI. It is concluded that p-Tau and EEG may provide biomarkers for olfactory system dysfunction also in humans post TBI.

Specific points

1. While the study design and experimentation are well versed and comprehensive, the mouse model used require attention in the clinical context, however. Namely, perhaps the most common injury to human olfactory system associated with TBI concerns shearing of olfactory nerve rather than direct trauma to the forebrain. This may concern also former NFL players who are used as reference in the Discussion section. Clearly, more justification is required for the mouse model in regard to the clinical relevance of the study.

2. Page 1, lines25-26, ‘examinations’ in plural on both lines.

3. Page 1, line 44. TBI is not ‘a leading cause of mortality worldwide’ according to the WHO statistics, it may cause significant ‘morbidity’. Revise.

4. Page 2, lines 65-67. The claim that ‘MRI scans..in TBI patients …are normal’, is incorrect. There are several advanced MRI techniques revealing abnormalities in brain already in hyper-acute phase. Text to be updated.

5. Page 5, line 233. ‘kruskkalwallis()’, is something mission from the parenthesis?

6. Page 10, Table 1 and page 12, Table 2. Are the p-values corrected for multiple comparisons? If not, Bonferroni correction or alike is needed.

7. Discussion section. There is a significant amount of repetition from results section, please remove overlap and prune the Discussion accordingly

8. List of references. Several journal names are in full, some abbreviated. Adjust bibliographic data as per JCM style throughout.

Author Response

March 29, 2022

Dr. Bruno Meloni 

Section Editor-in-Chief

Biomedicines

Manuscript ID: biomedicines-1649961

We are delighted to submit our revised manuscript (biomedicines-1649961) entitled Increases of phosphorylated Tau (Ser202/Thr205) in the olfactory regions are associated with impaired EEG and olfactory behavior in traumatic brain injury mice by Yoon et al.

We greatly appreciate the thoughtful and constructive comments from the Reviewers. As suggested by the Reviewer 1 and 2, we fully addressed and provided a point-by-point response to the comments by Reviewer 1 and 2. In addition, we performed a new experiment and provided new data (Figure 2). Changes to the text are highlighted (yellow) in the revised manuscript.

Response to  Reviewer 1

We thank to the Reviewer #1 for her/his constructive comments. We fully addressed comments as identified below:

  1. While the study design and experimentation are well versed and comprehensive, the mouse model used require attention in the clinical context, however. Namely, perhaps the most common injury to human olfactory system associated with TBI concerns shearing of olfactory nerve rather than direct trauma to the forebrain. This may concern also former NFL players who are used as reference in the Discussion section. Clearly, more justification is required for the mouse model in regard to the clinical relevance of the study. Response 1: As suggested by the Reviewer#1, we justified how the weight-drop TBI model in this study shows the clinical relevance to sports-related TBI in humans. Additionally, we performed a new experiment and checked axonal shearing in the olfactory-related regions by detecting amyloid precursor protein (APP), that is known as a axonal injury marker in human and animal of TBI. We added a new Figure #2 in page 6-7. This data shows that our TBI model is produce axonal injury for dysfunctions in olfactory bulb-associated regions.

We added a new paragraph about the clinical relevance of the current study into the Discussion as identified below (in Page 16, line 454-461):

“Mild repetitive TBIs are founded in boxers, foot-ball players and wrestlers those who may have axonal and cytoskeletal alterations and tauopathy in the brain [3]. Indeed, these types of TBI patients and animals models exhibited a strong diffuse axonal injury and the patients have symptoms of loss of consciousness, anxiety, fatigue, headache and concussive convulsion and/or impact seizure [4]. Mild TBI patients also have olfactory dysfunctions such as anosmia, the total loss of the sense of smell [5,6]. Our repetitive TBI mice showed a significant change of olfactory dysfunction-related behaviors compared to control mice.  Both the axonal damage in the olfactory bulb-associated areas and the loss of olfactory function-associated behaviors in our TBI animal model provide the clinical relevance of olfactory dysfunction in human TBI patients. Thus, this TBI animal model can be useful for further validating a precise mechanism of the olfactory dysfunctions of mild TBI patients [7]. 

We also added or replaced references appropriately in the revised manuscript.

  1. Page 1, lines25-26, ‘examinations’ in plural on both lines.

Response 2: As pointing out by the Reviewer #1, we replaced examination to examinations in page 1, line 26-27.

  1. Page 1, line 44. TBI is not ‘a leading cause of mortality worldwide’ according to the WHO statistics, it may cause significant ‘morbidity’. Revise.

Response 3: Thank you for pointing out the discrepancy in the term of mortality. We replaced the term “mortality” to “morbidity in line page2, line 46.”

  1. Page 2, lines 65-67. The claim that ‘MRI scans..in TBI patients …are normal’, is incorrect. There are several advanced MRI techniques revealing abnormalities in brain already in hyper-acute phase. Text to be updated.

Response 4: We appreciate the Reviewer#1 point on the lack of the background of recent MRI techniques for detecting of TBI status [8-11]. We rephrased the paragraph to information of several advanced MRI techniques in Page 2, line 65-69 as identified below:

“Recent studies show that advanced human imaging tools, such as MRI and CT scans, can detect pathology and prognostication in various brain regions of mild TBI patients  [8,12]. The state-of-the art imaging methodology even can determine pathology and prognostication within 72hr after TBI rather than 2 to 3weeks.”

  1. Page 5, line 233. ‘kruskkalwallis()’, is something mission from the parenthesis?

Response 5: The parenthesis was inserted to denote kruskalwallis as a function. In revision, we erased it on page 5, line 247.

  1. Page 10, Table 1 and page 12, Table 2. Are the p-values corrected for multiple comparisons? If not, Bonferroni correction or alike is needed.

Response 6: We did not perform the multiple comparisons for Table 1 and Table 2. The spectral powers (delta, theta, beta, low and high gamma) and the brain regions (frontal, parietal, and OB) are independent experimental conditions, no need for pairwise means between different pairs.

  1. Discussion section. There is a significant amount of repetition from results section, please remove overlap and prune the Discussion accordingly.

 Response 7: Thank you for your suggestion. The discussion section has been modified and pruned by removing overlapping content from the results section.

  1. List of references. Several journal names are in full, some abbreviated. Adjust bibliographic data as per JCM style throughout.

Response 8: The bibliographic data have been reformatted to compile with the MDPI citation style and the references list has been re-numbered in order of appearance in text on page 19-21.

We would be grateful if our revision is favorably looked upon and considered suitable for publication in Biomedicines.

Thank you very much for your deep consideration.

Sincerely yours,

Hoon Ryu, Ph.D.

Director

Laboratory for Brain Gene Regulation and Epigenetics

Brain Science Institute

Korea Institute Science and Technology

Reviewer 2 Report

Interesting paper looking at increased tau phosphorylation in olfactory region following TBI. 

Introduction: authors adequately point out early disruption in the olfactory system following TBI. This is primarily due to olfactory shearing. 

Methods: please justify why a weight drop model was used instead of a model that would produce more of the acceleration/deceleration components necessary for axonal shearing. 

Figure 1: the behavioral data is sufficient. In the discussion, additional experiments regarding fear conditioning (which can be disrupted with olfactory lesions) should be discussed. 

Figure 2: interesting findings and likely early formation of oligomers. Should be expanded upon in discussion with reference provided. 

Figure 3: logical as tau pathology is not due to change in concentration but change in morphology and function. 

Figure 4: it would be helpful here to stain for axonal bulbs as well. Again probably not as robust as if using an acceleration/deceleration model but should see some with weight drop and would be of interest to correlate. 

Figure 5: data is interesting. Please speculate into the role of inducing astrogliosis and potential seizure focus with the reference provided below. 

Figure 6: valid assays and interesting data. 

Both tables are sufficient and useful. 

Figure 7: please expand in discussion how this early behavior alteration can have potential lasting impacts for patients suffering from TBI. 

As mentioned above, please expand some important concepts including the role of seizure generating focus PMID: 35035475.

Overall the paper is well conducted and with the addition of the above information would be of interest to the readership. 

Author Response

March 29, 2022

Dr. Bruno Meloni 

Section Editor-in-Chief

Biomedicines

Manuscript ID: biomedicines-1649961

We are delighted to submit our revised manuscript (biomedicines-1649961) entitled Increases of phosphorylated Tau (Ser202/Thr205) in the olfactory regions are associated with impaired EEG and olfactory behavior in traumatic brain injury mice by Yoon et al.

We greatly appreciate the thoughtful and constructive comments from the Reviewers. As suggested by the Reviewer 1 and 2, we fully addressed and provided a point-by-point response to the comments by Reviewer 1 and 2. In addition, we performed a new experiment and provided new data (Figure 2). Changes to the text are highlighted (yellow) in the revised manuscript. We added and replaced references appropriately in the revsion.

Response to  Reviewer 2

We greatly thank to the Reviewer #2 for her/his encouraging comments for our manuscript such as “Overall the paper is well conducted and with the addition of the above information would be of interest to the readership.”

We addressed all comments suggested by the Reviewer #2 as identified below:

  1. Methods: please justify why a weight drop model was used instead of a model that would produce more of the acceleration/deceleration components necessary for axonal shearing. 

Response  1: As suggested by Reviewer#2, we justified our weight drop TBI mouse model as identified below (page 2, line 73 to 86):

“Past TBI models have utilized invasive methods such as controlled cortical injury (CCI), blast-induced TBI (bTBI), fluid percussion injury (FPI), and closed/head impact model of engineered rotational acceleration (CHIMERA). In this study, we produced TBI mice with repetitive impacts by weigh drop on head of mice to emulate human sub-TBI scenarios and examined the various neurological consequences of repetitive head injuries. Although the buried food-seeking test is the most reliable olfactory assay in rodents so far [18,19], it is reported to be closely associated with other cognitive deficits, such as abnormalities in food motivation or elevated fear. For this reason, it is hard to detect and measure subtle olfactory impairments may remain undetected [20]. To overcome this weakness, we used electroencephalogram (EEG) to measure neuronal oscillations of mice during an olfactory oddball paradigm test to scale olfactory functionality instead. Then, to avoid disruption from surgery, a separate mouse cohort was used to examine the neurological consequences of the olfactory system in repetitive TBI mice. “

Additionally, we performed a new experiment and checked axonal shearing in the olfactory-related regions by detecting amyloid precursor protein (APP), that is known as a axonal injury marker in human and animal of TBI [1,2]. We added a new Figure #2 in page 6-7. This data shows that our TBI model is produce axonal injury for dysfunctions in the olfactory bulb-associated regions. 

  1. Figure 1: the behavioral data is sufficient. In the Discussion, additional experiments regarding fear conditioning (which can be disrupted with olfactory lesions) should be discussed. 

Response 2: As suggested by Reviewer#2, we added further description about the fear conditioning-related behavior test into the discussion on Page 17, line 487-493 as identified below:

“We performed the buried food seeking test to evaluate olfactory function in the TBI models (Figure 8). It also showed significant changes in latency and duration for seeking buried food in TBI mice. But, we can also use other behavioral test such as olfactory-cued fear conditioning which detects freezing and fear-potentiated startle associated with the odor cue and electrical foot shocks[3]. It is a good animal model for examining the olfactory region and amygdala circuits effectively [4]. We are looking forward to use this behavior model for the future study.”

  1. Figure 2: interesting findings and likely early formation of oligomers. Should be expanded upon in discussion with reference provided. 

Response 3: As suggested by Reviewer#2, we added the more discussion on the early formation of Tua oligomers on Page 16, line 462-471 as identified below:

“It has shown that mild concussive TBI patients exhibit acute axonal accumulations of p-Tau within hours to weeks [5,6]. TBI-induced tauopathy in mouse and human are observed in both axonal and subcellular compartments through various molecular mechanisms such as dysregulation of kinases and protein phosphatases [7]. Furthermore, recent clinical study has demonstrated that plasma p-Tau levels and p-Tau/total Tau (T-tau) ratio in acute and chronic TBI patients are the better diagnostic and prognostic biomarkers than only T-tau level [8]. These phenomena are also observed in the mild repetitive TBI mouse model [9]. Taken together, our study suggests that p-Tau level and p-Tau/T-Tau ratio is also a neuropathological biomarker for the olfactory bulb-associated regions in mild repetitive TBI mouse model.”

  1. Figure 4: it would be helpful here to stain for axonal bulbs as well. Again probably not as robust as if using an acceleration/deceleration model but should see some with weight drop and would be of interest to correlate. 

Response 4: We appreciate the Reviewer#1’s suggestion on the staining of axonal bulbs.  Accordingly, we checked APP staining in the olfactory bulb-associated regions of TBI mice in Figure 2 on page 6-7. As a result, we confirmed that our TBI model also produced to the axonal damage in olfactory tract regions as shown below:  

  1. Figure 5: data is interesting. Please speculate into the role of inducing astrogliosis and potential seizure focus with the reference provided below. 

Response 5: We appreciate the reviewer for introducing a relevant and interesting paper. Our TBI-EEG data do not supply any clue about the post-traumatic epilepsy, yet the reference directly shows how the cellular level changes can be developed into neurophysiological dysfunctions. Hence we cited the paper in the Discussion as identified below (page 16, line 476-479):

“In the countercoup brain injury model, Small et al. reported that TBI induced the dysregulation of glutamate dysfunction in reactive astrocyte, resulting in an increased susceptibility to kainic acid-induced seizures [32].”

  1. Figure 7: please expand in discussion how this early behavior alteration can have potential lasting impacts for patients suffering from TBI. 

Response 6: As suggested by Reviewer#2, we added the more discussion about how the early behavior alteration affects TBI patient’s life as identified below (page 16, line 443-449):

“It is known that olfactory dysfunctions of TBI patients significantly decrease quality of life (QoL) score [10]. And olfactory dysfunctions are observed in the early phase (24hr) of mild TBI patients who have post-concussion and anxious symptoms at long-term phase (1 year) [11]. Our current data suggest that mild repetitive TBI mice show impairments of the olfactory-related behavioral symptoms (Figure 8), and this TBI mouse model is a good tool to find the mechanisms of olfactory dysfunctions in both acute and long-term phase of mild TBI patients.”

  1. As mentioned above, please expand some important concepts including the role of seizure generating focus PMID: 35035475.

Response 7: Thank you for the constructive comment, We discussed more about the suggested reference on page 16, line 476-479.

We would be grateful if our revision is favorably looked upon and considered suitable for publication in Biomedicines.

Thank you very much for your deep consideration.

Sincerely yours,

Hoon Ryu, Ph.D.

Director

Laboratory for Brain Gene Regulation and Epigenetics

Brain Science Institute

Korea Institute Science and Technology

Round 2

Reviewer 2 Report

Accept.